# A qualitative exploration of using financial incentives to improve vaccination uptake via consent form return in female adolescents in London

**Lauren Rockliffe, Selma Stearns, Alice S. Forster** *

Research Department of Behavioural Science and Health, UCL, London, United Kingdom

* alice.forster@ucl.ac.uk

**Data Availability Statement:** All datasets for this paper are available from the UCL data repository (https://rdr.ucl.ac.uk/articles/dataset/Acceptability_of_incentives/12659792).

## Abstract

### Objectives

Incentivising vaccine consent form return may improve vaccine uptake and be seen as less coercive than incentivising vaccination itself. We assessed the acceptability of financial incentives in this context among adolescent females and explored potential mechanisms by which incentives might change behaviour.

### Design

Focus groups and analysis of free-text questionnaire responses.

### Methods

Study 1: 36 female secondary students in London (age 13–14) participated in six focus groups exploring the use of incentives in the context of vaccination. Data were analysed thematically. Study 2: was conducted to triangulate the findings of Study 1, by analysing free-text questionnaire responses from 181 female secondary students in London (age 12–13) reporting their opinion of incentivising consent form return. Data from Study 1 was also used to explore perceived potential mechanisms of action by which incentives might encourage consent form return.

### Results

Focus group participants had positive attitudes towards incentives, with 61% of free-text responses also expressing this. Most focus group participants thought that incentives would encourage consent form return (18% of free-text respondents spontaneously also mentioned this). While incentivising consent form return was seen as ethical, focus group participants who incorrectly thought that vaccine receipt was being incentivised raised concerns about bribery, although only 4% of free text respondents reported these concerns. Frequently raised mechanisms of action included incentives increasing engagement with, and the perceived value of consent form return.

**Funding:** This work was funded by Cancer Research UK [C49896/A17429]. https://www.cancerresearchuk.org/ The funder had no role in study design, data collection and analysis, decision to publish, or preparation of the manuscript.

**Competing interests:** The authors have declared that no competing interests exist.

## Conclusions

Adolescents had positive views of financially incentivising consent form return to promote vaccine uptake, although care must be taken to reduce misconceptions regarding what is being incentivised. Incentivising vaccination was seen as coercive, but incentivising actions that increase the likelihood of vaccination (i.e. consent form return) were not. Incentives may encourage adolescents to return consent forms by helping them engage with this behaviour or increasing its' perceived value.

## Introduction

In the UK, vaccines for school-aged children are mainly offered through school-based programmes, and similar approaches are used in Australasia, Europe and North America for some vaccines. In school-based vaccination programmes in the UK, consent for vaccination is sought, most commonly, using paper forms which students are required to get signed by a parent or guardian and return to school by hand. Parents are asked to indicate on the forms whether consent is being given or declined. Unpublished data from health authorities consistently show that a large proportion of consent forms are not returned; for example around 45% of child flu vaccine consent forms are unreturned (personal communication–Dr Heffernan, NHS England (London)). In most instances, vaccinations will not be given where consent forms are unreturned. Evidence suggests that returned forms are more likely to provide consent than decline it [1], thus motivating consent form return may increase vaccination uptake [2].

Presently, the return of consent forms is equally as mandatory as the return of other forms in schools. On a school-by-school basis, young people may receive punishment for not returning forms (a form of disincentive) and immunisation nurses also spend a large amount of time chasing adolescents and their parents to encourage them to return their consent forms. However, consent form return remains inadequate. There is increasing evidence that behavioural incentives are a useful approach to increase consent form return [1, 3], particularly for adolescent vaccinations, where young people play an increasingly important role in facilitating consent; in the UK these are human papillomavirus (HPV), MenACWY protecting against meningitis types A, C, W and Y, and the 3-in-1 teenage booster protecting against tetanus, diphtheria and polio. Behavioural incentives are the offer of material or non-material rewards for attaining a goal [4]. They have been used as a tool to change health-related behaviour in a number of contexts [5].

Incentives have been used widely in the context of improving vaccination uptake, with varying effects [6, 7]. Parents and healthcare providers have often been the target of incentives to improve childhood vaccine uptake, with a focus on infant and pre-school immunisations. An increasing number of vaccines are available for adolescents, however fewer studies have considered incentives aimed at this group. Mantzari and colleagues found improvements in HPV vaccine uptake following the offer of a financial incentive, with 22% of 17–18 year old girls who were offered the incentive completing the vaccine services compared to 12% of the usual care arm [8]. Forster et al [1] explored the effect of incentivising 12–13 year olds to return a vaccine consent form signed by their parent, offering them entry in a prize draw to win a shopping voucher as part of a cluster randomised feasibility study (n = 575). Adolescents were eligible for prize-draw entry regardless of whether the returned consent form was

providing or withholding consent to vaccination. Around 76% of those randomised to the incentive arm of the study received the vaccine compared to 61% of adolescent in the control arm (it was not appropriate to test for significance in this feasibility study, a larger study is required to explore efficacy). Similar findings were reported when school classes and individual adolescents were incentivised to return Hepatitis B vaccination consent forms [3].

There are several plausible mechanisms for incentives changing behaviour [9], many of which suggest that the impact of incentives could be greater among adolescents than among adults due to differences in reward-related cognition and social cognition [10] (Box 1). Expectancy-value models suggest that incentives increase the perceived value of carrying out the behaviour, which in turn increase intentions to engage in the target behaviour [9]. Relatedly,

## Box 1. Plausible mechanisms of action of incentives

### Cognitive (+ social) mechanisms[a]

| | |
|---|---|
| Expectancy-value | Incentive increases perceived value of target behaviour |
| Temporal discounting | Incentive hastens receipt of reward |

### Social mechanisms

| | |
|---|---|
| Fear of missing out | Individual fears missing out on experience their peers are having. Relevant when the incentive is an experience |
| Anticipated regret | Individual anticipates that they will regret not engaging with the target behaviour if a peer were to get the incentive but the individual did not |

### Memory-based mechanism

| | |
|---|---|
| Enhanced memory | Rewards enhance memory to engage with the behaviour |

[a] *Heightened in social settings where peers prime a reward-sensitive motivational state. Particularly pertinent among adolescents.*

incentives hasten the speed with which individuals are rewarded for their health protective behaviour (immediate receipt of incentive versus longer term disease prevention; temporal discounting models). This may be particularly pertinent for adolescents who typically have a greater preference for choices that have an immediate reward, although there is individual variability in this propensity to temporally discount rewards [11]. This effect may be heightened if the incentive is offered in a social setting, where peers prime a 'reward-sensitive' motivational state [12], which may further increase the value of rewards that are immediately available. Adolescents are particularly sensitive to this peer influence [12]. Peers may also mediate the effect of incentives in scenarios where the incentive itself is an experience, by causing individuals to express fear of missing out on a rewarding experience that one's peers are having [13]. For example, entry into a prize draw may be viewed as an exciting opportunity and this in turn may encourage individuals to engage with the behaviour required to obtain the incentive. Closely related to this, individuals may anticipate that they will feel regretful if a peer was to receive the incentive but they did not, motivating engagement with the behaviour. Finally, rewards have been linked to improved memory, so it is possible that incentives increase the likelihood that the individual will remember to perform the behaviour [14]. While these mechanisms of action are theoretically plausible to explain any effect of incentives on adolescents' health behaviour, they have only rarely been explored in this population [15, 16], and not in the context of infrequently performed health behaviours such as vaccination for which the mechanisms of effect are likely to be different.

In order to adopt incentivising adolescents as an approach to increase vaccine uptake, the approach must be deemed acceptable. Acceptability is "a multi-faceted construct that reflects the extent to which people delivering or receiving a healthcare intervention consider it to be appropriate, based on anticipated or experienced cognitive and emotional responses to the intervention" and is a necessary but not sufficient condition for intervention efficacy [17]. Evidence suggests that the use of some forms of non-financial incentives is deemed acceptable to parents and health professionals [6], however, the same cannot always be said for approaches using financial rewards. Financial incentives are sometimes viewed by adults as a form of bribery or coercion, or as undermining individual autonomy or being unfair or inappropriate [18, 19]. Financially incentivising actions that increase the likelihood of the target behaviour may be perceived as less coercive (e.g. consent form return). Despite understanding how adults view the use of financial incentives, we do not know how adolescents view such interventions which is crucial for the development of effective and acceptable incentive interventions aimed at this population.

In the present research, we explored the acceptability of financial incentives directed at adolescents via two studies that were conducted as part of a process evaluation of the cluster randomised controlled feasibility trial, described above, that incentivised HPV vaccination consent form return as a means of improving vaccination uptake [1, 20]. The objectives of this paper are to 1) assess the acceptability of financial incentives to promote vaccine consent form return among adolescents and 2) explore the potential mechanisms by which financial incentives might change behaviour amongst this group.

## Material and methods

### Context

Participants for both studies had previously participated in the feasibility trial of an intervention to increase HPV vaccine uptake by incentivising vaccine consent form return (described above). The trial protocol was registered (ISRCTN72136061) and published [20]. In brief, participants were 12–13 year old female students (N = 575) from 6 secondary schools in two

London Boroughs. Uptake of Dose 1 of the HPV vaccine for the participating Boroughs in the year preceding the trial was 82% and 90% respectively, compared to the England average of 87%. We approached all secondary schools in participating boroughs initially via email and then by telephone. We stopped recruitment once a sufficient number of schools agreed to participate. Schools were randomised to either the usual care or incentive intervention arm. The incentive was the opportunity to be entered into a prize draw to win a £50 ($65 / 60 euro) shopping voucher, with a 1 in 10 chance of winning (herein referred to as £50-1/10), if they returned a completed HPV vaccine consent form (regardless of whether consent was given or withheld).

## The present studies

Data were collected from two studies: Study 1 comprised focus groups with adolescent girls and explored the acceptability of incentivising HPV vaccination consent form return following their participation in the trial. Study 2 used free text responses provided by a large group of adolescent girls collected as part of the trial, regarding their attitudes towards incentivising HPV vaccine consent form return. These data were used to triangulate the findings of Study 1, using different methods. Some individuals may have participated in both studies, but data were collected anonymously so it is not possible to determine how often this occurred. At the time of study, the HPV vaccine was only available for females, but a gender-neutral programme now runs in the UK.

Study 1 was conducted between March and April 2018 and Study 2 between July 2016 and January 2017, both in London, UK. Ethical approval for both studies was granted by UCL Research Ethics Committee (7427/005 and 6615/002).

## Sample and recruitment

**Study 1.** Participants were Year 9 female students (aged 13–14) recruited from two secondary schools in London. All Year 9 female students who had previously participated in the feasibility trial (when they were 12–13 years) were eligible to participate. Opt-out consent was requested from parents for the study, and students were made aware of the opportunity to participate in the study by a member of school staff. All eligible participants were provided with an information sheet prior to participating and written assent was gained.

**Study 2.** Participants were Year 8 female students (age 12–13) recruited from three secondary schools as part of the feasibility trial [1, 20]. All had been offered the incentive within the last four weeks. Students and their parents received an information sheet prior to participating. Opt-out consent was requested from parents for the study and assent from the student was assumed based on completion of the questionnaire used for data collection.

## Data collection

**Study 1.** Six focus groups were conducted. We chose to use focus groups to elicit discussion between participants and to provide a less intimating setting for girls to participate. Focus groups were conducted in the students' respective schools, during the school day. Groups comprised between five and eight participants (an average of six per group), all of whom were familiar with one another and were facilitated by two of four female researchers. One had a PhD and three had MSc qualifications; all had conducted focus groups previously. Participants did not have a relationship with the researchers prior to the focus groups, who were told that they were researchers from a university. No other people were present during the groups.

A topic guide was used to direct the discussion, exploring participants' experience of being offered the £50-1/10 incentive in the previous trial, attitudes towards use of incentives in

the context of vaccination in general, and preferences for the nature of the incentive. The topic guide was developed by the research team. It was informed by existing research but was not driven by a particular theory. In addition to exploring experiences of the £50-1/10 incentive, participants were specifically questioned about two alternative incentives: 1) every person is offered £3 ($4 / 3.5 euro) if they returned the consent form (herein referred to as £3-all) and 2) individuals are offered entry into a prize draw to win a £300 ($400 / 350 euro) shopping voucher with one winner if they return the consent form (herein referred to as £300 prize draw). For each of these options participants were shown visual representations of the odds using icon arrays. Participants were not asked explicitly about potential mechanisms of action. All focus groups were audio-recorded, transcribed verbatim and lasted an average of 29 minutes (ranging from 23 to 35 minutes). Transcripts were not returned to participants for correction / comment. Field notes were made during the focus groups but were not analysed.

All participants completed a demographic questionnaire that assessed deprivation using the Family Affluence Scale [21] and ethnicity using UK census categories [22].

**Study 2.**   Within a longer questionnaire (see S1 File), participants in Study 2 were asked to respond to the question *"What did you think about being entered into a prize draw to win a £50 voucher if you returned the HPV vaccine consent form?".* Participants could provide multiple opinions, and all written responses were included in analysis. Demographic data collected via the questionnaire included religion (based on Office for National Statistics, 2011, [22]) and migration status (whether they and their parents were born in the UK; adapted from Marlow et al. 2015 [23]). Questionnaires were distributed and collected by the students' form tutors and completed during a school lesson.

## Analysis

We used Braun and Clarke's phases of thematic analysis to guide analysis of data from Study 1, which was conducted by two researchers (author 3 and author 1) [24]. Both researchers read the raw data and identified codes to apply to the data. Following discussion regarding the codes, author 1 applied an agreed coding frame to the data, using the qualitative data analysis software Nvivo 12. All coding was reviewed by author 3 and refinements then made to the coding frame by author 1 and author 3. Disagreements for coded data were resolved through discussion between both coders.

The first objective of this paper was to assess the acceptability of financial incentives in the context of vaccination among adolescents. In line with deductive thematic analysis, we used Sekhon et al.'s framework of acceptability as a guide to organising the codes into themes for this part of the analysis [17]. The framework consists of seven component constructs including affective attitude (how the individual feels about the intervention), burden (the perceived amount of effort required to participate in the intervention), perceived effectiveness, ethicality (how well the intervention fits with the individual's values), intervention coherence (how well the individual understands the intervention), opportunity costs (the extent to which benefits must be given up to participate), and self-efficacy. Codes that did not fit this structure were noted and new themes generated. Responses pertaining to the second objective (exploring mechanisms of action) were organised into themes with no hypothesis *a priori*. Then, in line with 'axial coding' [25], we considered how codes were related, which also helped to establish themes and our reporting of them.

Finally, two researchers (author 3 and author 2) independently applied the adapted coding/ thematic framework to the free-text data generated in Study 2. Data that did not fit the framework were noted and reported separately. There was some agreement on 86% of responses and all discrepancies were resolved by discussion.

The results present a summary of the themes identified within the data in Study 1 and the prevalence of these themes in Study 2. Illustrative quotes are presented with focus group and participant number or 'unknown' where it was not possible to identify who to ascribe the response to. Analysis of free text data is described, supplementing the results of the focus groups where a sizable proportion of respondents made a similar response or where findings from the two studies differed. The prevalence of all codes in free text data relating to the first objective are provided in Table 2 in the result section. It was clear that participants had conflicting opinions regarding some of the incentive options (for example, believing that the £300 prize draw would be wasteful, but also motivational). For this reason, Table 3 provides a summary of focus group participant responses relating to each of the three incentive options, discussed by theme; £50-1/10, £3-all, and £300 prize draw. This manuscript adheres to the COREQ guidelines [26].

## Results

### Sample characteristics

**Study 1.** No parent opted their daughter out of Study 1 and all adolescents invited to participate assented to do so. A total of 36 adolescents participated, 37% of whom reported being from a White British ethnic background and 67% of whom came from highly affluent families (Table 1). Two participants had won the shopping voucher in the previous trial and all had been offered the chance to receive it. Two participants had not returned their HPV vaccine consent form and eight had received no doses of the vaccine.

**Study 2.** There were 16 parents (3% of those eligible) who opted their daughter out of the study. Around 80% of adolescents invited to complete a questionnaire did so. Free text responses were obtained for 181 participants (89% of those who completed a questionnaire; 71% of those invited to complete the questionnaire). Christianity was reported by half of participants (50%), and most participants were born in the UK (91%). Around 93% of those providing a response had returned their HPV vaccine consent form and 89% had received the first dose of the vaccine.

### Objective 1: Acceptability of incentivising vaccine consent form return

Adolescents' views of the acceptability of incentivising vaccine consent form return span five unique themes. See Table 2 for the prevalence of all codes in free text data relating to this first objective, and Table 3 for a summary of focus group participant responses relating to each of the three incentive options, discussed by theme; £50-1/10, £3-all, and £300 prize draw.

**Affective responses to incentives.** Participants in Study 1 were generally positive about the use of the £50-1/10 incentive, with many describing it as a *"good idea"* that made the vaccination experience slightly more positive. One participant explained that being given the opportunity to receive the £50-1/10 incentive made her feel special. Many participants described their desire to win, and their excitement at the prospect.

*"I like it. Because it was just quite exciting. Because you don't get those opportunities ever, really. So, it was quite exciting"*

*(FG1, P1)*

General positivity about incentives was the most common response provided in Study 2 (61%, 111/181) and was often coupled with sentiments indicating a belief that the £50-1/10 incentive would motivate consent form return.

**Table 1. Demographic characteristics of participants in Study 1 and 2, and the feasibility trial from which respondents in both studies had participated.**

| | Feasibility trial N = 575[a] | Study 1 N = 36 | Study 2 N = 181[a] |
|---|---|---|---|
| **Ethnicity *n* (%)** | | | |
| White British | 39 (41) | 14 (37) | 25 (51) |
| African | 21 (22) | 0 (0) | 8 (16) |
| White Other | 6 (6) | 2 (0.6) | 4 (8) |
| Other | 28 (30) | 20 (56) | 12 (25) |
| **Religion *n* (%)** | | | |
| None | 70 (18) | - | 62 (34) |
| Christian | 259 (67) | - | 91 (51) |
| Muslim | 39 (10) | - | 18 (10) |
| Other | 19 (5) | - | 9 (5) |
| **IMD quintile *n* (%)** | | | |
| Most deprived: 1 | 380 (66) | - | 96 (53) |
| 2 | 107 (19) | - | 31 (17) |
| 3 | 46 (8) | - | 23 (13) |
| 4 | 21 (4) | - | 14 (8) |
| Least deprived: 5 | 21 (4) | - | 17 (9) |
| **Family affluence scale *n* (%)** | | | |
| High affluence | - | 24 (67) | - |
| Medium | - | 12 (33) | - |
| Low affluence | - | 0 (0) | - |
| **Migration status *n* (%)** | | | |
| Girl born UK, parents born UK | 83 (23) | - | 51 (31) |
| Girl born UK, 1 parent born UK | 40 (11) | - | 21 (13) |
| Girl born UK, neither parent born UK | 188 (53) | - | 75 (46) |
| Girl not born UK, neither parent born UK | 44 (12) | - | 14 (9) |
| Girl not born UK, 1 parent born UK | 1 (0.2) | - | 1 (0.7) |
| Girl not born UK, parents born UK | 1 (0.2) | - | 1 (0.7) |
| **Had 1st dose of HPV vaccine *n* (%)** | 391 (89) | 28 (78) | 151 (89) |

[a] Total not equal to N due to missing data.

- indicates that data were not collected.

A number of participants in Study 1 felt that being offered the £50-1/10 incentive was an unusual and interesting opportunity that they would not normally be given. Some of these participants commented that the type of incentive offered (£50 shopping voucher) was relevant to girls of their age, which was perceived to be a positive thing.

However, participants in Study 1 felt that there was the potential for some girls to experience negative feelings should they not be in receipt of the prize. A number of these participants were concerned that not winning might make some girls *"feel left out"*, cause disappointment or annoyance, or might even discourage them from returning a consent form in the future (very few respondents in Study 2 expressed these concerns (3%, 5/181)). Furthermore, some participants in Study 1 felt that offering an incentive may cause jealousy, especially if the value of the prize was high and/or there was only one winner (a view not expressed by respondents in Study 2). These issues were all discussed in the context of 'other girls' and not the participants themselves.

**Table 2. Counts/percentages of themes, codes and sub-codes derived from free text comments in Study 2.**

| Themes, codes and sub-codes | n | % | Themes, codes and sub-codes | n | % |
|---|---|---|---|---|---|
| **Affective responses to incentives** | | | **Inclusivity** | | |
| Generally positive | 111 | 61.3 | Open to all | 0 | 0.0 |
| Unusual | 0 | 0.0 | Not dependent on parents | 0 | 0.0 |
| Relevant to demographic | 6 | 3.3 | Dependent on parents | 0 | 0.0 |
| Cause disappointment if don't win | 3 | 1.7 | Involves the students | 1 | 0.6 |
| Discourage future engagement with consent form return | 0 | 0.0 | | | |
| Jealousy | 0 | 0.0 | **Incentive-specific attitudes** | | |
| Forgot about it | 2 | 1.1 | *Attitude towards the prize* | | |
| Indifferent | 32 | 17.7 | Good prize | 32 | 17.7 |
| Not motivational[a] | 8 | 4.4 | Bad prize[a] | 2 | 1.1 |
| Didn't know about the prize[a] | 5 | 2.8 | Unbelievable | 2 | 1.1 |
| | | | *Attitude towards the odds of winning* | | |
| **Ethicality of incentives** | | | Good odds | 2 | 1.1 |
| Bribery | 8 | 4.4 | Bad odds | 11 | 6.1 |
| Fair | 0 | 0.0 | | | |
| Unfair | 7 | 3.9 | **Likelihood of behaviour change** | | |
| *Inappropriate* | | | Incentives are effective | 33 | 18.2 |
| Wasteful | 0 | 0.0 | Would return consent form anyway | 17 | 9.4 |
| Too much money for age group | 0 | 0.0 | Not high enough value prize to motivate change | 0 | 0.0 |
| Disproportionate to returning consent form | 2 | 1.1 | | | |
| Should promote vaccination instead | 3 | 1.7 | | | |

[a] Indicates additional code.

> *"Because if it's like one out of 100 people winning it that one person's winning it. And, if the money is quite low then they wouldn't be that like jealous. But if it's really high then there might be more chance of people getting annoyed about it"*
>
> *(FG5, P28)*

While many participants in Study 1 described being excited by the offer, some participants felt indifferent; and this was also expressed by respondents in Study 2 (18%, 32/181). Other participants in Study 1 claimed to have forgotten about the £50-1/10 incentive due to a belief that they would be unlikely to win, or because their main focus was on having the vaccination.

**The ethicality of incentives.** *Bribery.* A number of participants in Study 1 felt that the offer of the £50-1/10 incentive acted as a bribe for them to return the consent form or made them feel *"like you're getting paid for it"*, although this was almost exclusively reported by participants who believed that vaccine receipt was being incentivised (and not consent form return). However, not everyone agreed, and some participants felt that because it was a prize draw and not a guaranteed reward for every girl it did not feel like bribery.

> *"It's like one out of ten. So, like it's not they're not bribing you. It's just like something there for you to like, look forward to or something. I feel like that yeah. But I don't think it's bribing"*
>
> *(FG5, P27)*

**Table 3. Summary of focus group participant responses towards different incentive options (Study 1).**

| Incentive | Theme | Summarised participant responses |
|---|---|---|
| £50-1/10 | Incentive-specific attitudes | • Most participants happy with the prize value and prize itself (shopping voucher)<br>• Prize value perceived to be *"quite a lot of money"* |
| | Ethicality of incentives | • Prize draws, compared to guaranteed rewards for every girl, perceived to feel less like bribery for some participants<br>• Prize draws with a limited number of winners perceived to be fair by some participants, as all girls would have the chance to win<br>• Potential for winner to feel bad if were the only person to win or they did not receive the vaccination |
| | Perceived likelihood of behaviour change | • Majority of participants agreed that the incentive was effective |
| £3-all | Incentive-specific attitudes | • Some participants liked that the incentive would be received by everyone<br>• Low value felt to be *"better than nothing"* by some<br>• Prize value felt to be too low to purchase anything meaningful by some participants<br>• Low value prize felt to be unexciting and not special |
| | Ethicality of incentives | • Some felt a guaranteed prize for all girls would be the best approach<br>• Felt to be wasteful and "not necessarily a good way to spend £300" if 100 girls were to receive the incentive |
| | Perceived likelihood of behaviour change | • Felt by some that this would be an effective motivator<br>• Others did not feel it would be motivating<br>• Some felt it would not be as effective as the £50-1/10 incentive |
| £300 prize draw | Incentive-specific attitudes | • Participants were enthusiastic<br>• Value perceived to be a large sum of money |
| | Ethicality of incentives | • Some believed that splitting a sum of money this size amongst more people would be better, to provide more girls the opportunity to win<br>• Felt to be too much money for girls of their age, considering it had not been 'earnt' |
| | Perceived likelihood of behaviour change | • Felt to be an effective incentive due to larger prize value |

Furthermore, one participant commented that *"It's not like you're forcing us to [return the consent forms]"*. Bribery was rarely mentioned by respondents in Study 2 (4%, 8/181), but respondents who did mention it almost always misunderstood what was being incentivised.

*Fairness.* Some participants in Study 1 felt that a prize draw with a limited number of winners was fair, as all girls would have the chance to win. However, others felt this was not an appropriate way of distributing the money and that a guaranteed prize for all girls (e.g. £3-all) would be a better approach. Relatedly, many participants felt it would be better to split a £300 prize draw incentive among more people, to give more girls the opportunity to win. One participant commented that she would feel guilty receiving even a £50 prize if she was the only winner; another participant, who had previously won the £50-1/10 incentive, explained that she felt it was unfair that she won, as she did not have the vaccination. Respondents in Study 2 rarely expressed sentiments of fairness (0%) or unfairness (4%, 7/181).

> *"So, if you have a budget of 300, then you'd rather give it to more people. Like at least like 50 each person. If you're going to spend it all . . . 300 on one person, it's a bit . . . a lot"*
>
> *(FG4, unknown)*

*Inappropriateness.* Some participants in Study 1 felt that a £3-all incentive would be wasteful and *"not necessarily a good way to spend £300"* if 100 girls were to receive it, for example. However, some participants felt that a £300 prize draw incentive was too much money for someone of their age to win, especially considering that it had not been 'earned'. Other participants felt that the offer of any incentive was disproportionate compared to the simplicity of the behaviour being incentivised (returning a form) and one believed that promoting the benefits of the vaccination was more appropriate. These views were not often expressed in Study 2 (between 0 and 2% of responses).

> *"Like I feel like it's quite a lot of money to get without earning it in any way. I mean like you got a vaccination but it's still. . ."*
>
> *(FG3, P16)*

**Incentive-specific attitudes.** *Attitude towards the prize.* Most participants in Study 1 were happy with the value of the £50-1/10 incentive as they perceived it to be *"quite a lot of money"*, especially for girls of their age, and with the prize itself (a shopping voucher) described as a *"good prize"*. Many respondents in Study 2 also described the £50-1/10 incentive as a good prize (18%, 32/181), but not all (~1%). Participants in Study 1 also gave positive responses when asked about a £3-all prize; some participants liked that the incentive would be received by everyone and although it was a low value reward, a number of participants felt that it was *"better than nothing"*. When subsequently asked about a £300 prize draw participants were enthusiastic, as this was perceived to be a large sum of money for girls of their age. However, some participants stated that a £50 prize would still be sufficient to encourage them to return a consent form.

> *"I think it was a good prize [£50-1/10]. I mean, you can't expect much but it was a good prize. . . I think it was kind of motivating because, you know, you get shopping in return"*
>
> *(FG4, P23)*

Conversely, some participants in Study 1 felt that the value of a £3-all incentive would be too low to purchase anything meaningful. Others felt that the incentive would be unexciting and *"doesn't make it really that special"*.

For some participants in Study 1, the offer of the £50-1/10 incentive felt unbelievable and like a *"hoax"*. Participants explained that the high prize value and high odds of winning lead them to believe it was untrue. A lack of understanding as to why the incentive was being offered also led some participants to question the genuineness of the offer. A couple of participants explained that the higher the value of the prize, the less real the offer would seem, since competition prizes are normally lower in value.

> *"Because it's like, it's kind of offering a lot of money to a lot of people. It kind of just seems a bit odd. So, like, if it was like, £10 to the same amount of people. Or £50 to, like, a smaller amount of people then it might be more believable"*
>
> *(FG1, P3)*

*Attitude towards odds of winning.* While a small number of participants in Study 1 felt that having a 1 in 10 chance of winning a prize were good, or *"reasonable"* odds, the majority of participants felt that the likelihood of winning would be very low. Unsurprisingly, participants

had a similar opinion of incentives with lower odds of winning (e.g. the £300 prize draw) and some participants felt that very low odds might deter girls from returning their consent forms. Nonetheless, participants claimed to be very motivated by high prize values, even if the odds of winning remained very low. Around 6% of respondents in Study 2 reported that a 1 in 10 likelihood of winning was low (11/181).

*". . .and your chances are very low as well [£300 prize draw] [. . .] It'll make me want to bring it back but I'll still have . . . Like I'll still be like I'm never going to win that, so . . ."*

*(FG4, unknown)*

**Likelihood of behaviour change.** Most participants in Study 1 agreed that the £50-1/10 incentive was, or would be, effective at motivating girls to return their consent form. Many respondents in Study 2 also indicated that they thought the approach would motivate behaviour (18%, 33/181), although some expressed the opposite opinion (4%, 8/181). A couple of participants in Study 1 felt that a £3-all incentive would be an effective motivator, although others disagreed, preferring higher value prizes. Some participants stated that they would return the consent form anyway because they thought vaccination was important. Around 9% of respondents in Study 2 also expressed this (17/181). For these participants the incentive was not perceived to make much of a difference, although for some it was viewed as a *"thank you"*.

*"I thought it was good incentive, but I probably would have given the form back anyway"*

*(FG2, P10)*

**Inclusivity.** Participants in Study 1 commented on the fact that all girls were eligible to win the £50-1/10 incentive, regardless of their vaccination decision. The £50-1/10 incentive was described as *"open for everyone"*, which was perceived by some to be a positive aspect. Since vaccine decision-making often falls on parents, one participant commented that for the girls the incentive *"brings them into it a bit"*, which they felt was positive.

*"I guess it was motivation to give in your HPV vaccines [consent forms] but I think quite a lot of parents were just forcing us to do it anyway so. . ."*

*(FG3, P16)*

Sentiments of inclusivity was made by only one respondent in Study 2.

## Objective 2: Mechanisms of action

Participants in Study 1 spontaneously discussed four possible mechanisms by which incentivising consent form return could encourage consent form return (see Table 4 for illustrative quotes):

1. The incentive encourages engagement with the behaviour—Many of the participants talked about the way in which the incentive encouraged girls to engage with returning their consent form.

2. The incentive increases the perceived value of the behaviour—A number of participants spoke explicitly about *"getting something in return"* for returning their consent form and

**Table 4. Quotes illustrating the possible mechanisms of action of the incentive.**

| Mechanism | Illustrative quote (from Study 1 and Study 2) | Frequency mentioned in free text responses (Study 2) n (%) |
|---|---|---|
| Incentive encourages engagement with the behaviour | *"Yeah, it makes people like, think about what they could have if they think about the prize you get. And then you think about, like, what gets you to the prize? As, handing in the forms, say"* (FG1, P1) | 33 (18) |
| Incentive increases the perceived value of the behaviour | *"My friends thought it was a good idea because it would, like, make people bring it back and have like, the thought to want to bring it back. So, they have the opportunity to be able to win, sort of"* (FG1, P6) | 30 (17) |
| Incentive acts as a reminder to perform the behaviour | *"it just like makes you remember more to hand it back"* (FG4, P20) | 0 |
| Desire to gain incentive instead of peers | *"Yeah. It also kind of makes you want to get the voucher more because otherwise one of your friends will get the voucher"* (FG3, P18) | 0 |
| Increases the speed that target behaviour is performed | *"It made me return the form quicker"* (free-text respondent) | 5 (3) |

their desire to have the opportunity to win a prize, suggesting that returning a consent form held more value when an incentive was attached to it, than it may have done previously.

3. The incentive acts as a reminder to perform the behaviour—A number of participants felt that the incentive acted as a reminder to return the consent form.

4. Desire to gain incentive instead of peers—One participant expressed their desire to be included in the prize draw and explained that the potential for one of their friends to win the prize instead of them, acted as a motivator to return their consent form.

Mechanisms of action were not frequently mentioned in Study 2 (see Table 4). The most commonly suggested mechanisms were the incentive encouraging engagement with the behaviour (18%, 33/181) and that incentives increase the perceived value of the target behaviour (16%, 33/181). Participants also noted that the incentive increased the speed with which they returned their consent form (3%, 5/181), which was not raised by participants in Study 1.

## Discussion

Incentivising vaccine consent form return is likely to increase vaccine uptake [1, 3]. We report the findings of two qualitative studies with adolescent girls exploring the acceptability of financial incentives to promote vaccination consent form return and perceived potential mechanisms by which financial incentives might encourage consent form return among this group. The majority of respondents were positive about the intervention, finding it an exciting prospect and many believed it would be an effective approach to behaviour change. However, the ethicality of incentives was questioned by a minority. Respondents spontaneously reported mechanisms by which incentives might work, including that they increase engagement with, and the perceived value of returning consent forms.

While the majority considered incentivising consent form return to be an ethical approach, among those who misperceived that vaccination was being incentivised (rather than consent form return), concerns were raised regarding the intervention being a bribe for vaccination. Concerns regarding incentives being coercive have also previously been reported by adults,

with particular worry for vulnerable groups who may be less able to decline them [19, 27]. Our findings highlight the importance of identifying suitably acceptable targets for incentives. In this context, increasing vaccination uptake was not considered to be an acceptable target for incentives, whereas a proximal target, which was open to the wider population (consent form return) was considered appropriate. In many instances the primary focus for health promotion initiatives (in this case increasing vaccination uptake) will not be the most acceptable target for incentives; proximal targets that increase the likelihood of the target behaviour being performed (e.g. consent form return), which are open to a wider population, appear more appropriate.

Incentives were considered by most participants to be a positive approach to facilitating consent form return. They found the opportunity exciting and unusual, and many believed the offer of a reward had encouraged girls to return their consent forms. This adolescent population appears to be more amenable towards incentives than adults have been in previous work. For example, McNaughton et al. found parents to have 'overwhelmingly negative reactions' to financial incentives for vaccination [19]. Indeed younger adults have been shown to be more agreeable towards incentives than older adults [28]. The difference may be because the incentive and target behaviour examined here differed from previous work, or because adolescents are generally more open to this intervention. Incentives, whether offering the chance to win a large prize or the certainty of a smaller reward, were seen on the whole to be a fair intervention that was 'open to everyone'. Adults have also previously expressed that targeted incentives offered to a particular group were seen as less acceptable than those offered to the wider population [19, 28, 29]. Although there may be differences between adolescents and adults in their acceptance of incentives, the same types of concerns and benefits are raised by both groups (just to differing degrees) [30]. There was no clear preferred prize option and no option was without its critics, for example the £300 prize draw was seen as both motivational and unfair.

Our findings support several possible mechanisms of action that have been suggested previously [9, 14]. Many participants spoke of the incentive making them return a consent form, in a way that suggested that the perceived value of returning a form had increased, and some participants explicitly said this. An alternative mechanism previously raised in the literature is that incentives enhance memory, suggesting that individuals are more likely to remember to perform the behaviour in the presence of an incentive. Many focus group participants reiterated this suggestion, stating that it helped them to remember to return the consent form or to remind their parent to sign the form, although this was not mentioned in the free-text responses. We found little evidence to support the hypotheses that fear of missing out or anticipated regret may explain any effect of incentives, although one participant highlighted her desire to win the incentive over her peers. There was little reference to the influence of peers heightening any effect of the incentive in general, although this may be too subtle for adolescents to recognise and report. Indeed, it may be that only the most cognitively available mechanisms were discussed in the two studies. It will be important to further triangulate these findings using different methodologies.

*Limitations.* While participants were recruited across a range of schools, our study was conducted in only two areas of London; adolescents living elsewhere may have responded differently. Participants in the focus groups were familiar with one another, and while this may have helped to facilitate discussion, some participants may not have expressed themselves as openly as they might have done in a one-to-one situation or may have only provided socially desirable responses. It is possible that participants from families with lower incomes may have refrained from commenting on the monetary value of financial incentives, through fear of being seen as valuing small value incentives. However, most participants in this study were from families with low levels of socio-economic disadvantage, which is likely to have affected their opinions

about the monetary value of financial incentives in other ways. It would be interesting to compare our findings to a similar study using a cohort with a more mixed demographic.

## Implications

Given the increasing evidence of the efficacy of facilitating vaccine uptake by incentivising vaccine consent form return [1, 3], we now need to identify how to successfully implement such an intervention. Our findings indicate that this will be dependent on individuals having a clear understanding of what target behaviour is being incentivised and incentivising behaviours that are available to all in the eligible population (e.g. not just those who choose to vaccinate). This study provides evidence for the first time regarding the types of incentives that are valued by adolescents, which will be useful for other researchers considering similar interventions. Future research should test the potential mechanisms of action raised in this study using experimental medicine approaches [31].

## Conclusion

Financial incentives to promote vaccine uptake via improving consent form return are acceptable to adolescents in general. There were clear preferences for incentives aimed at proximal actions that increase the likelihood of vaccination, such as consent form return, rather than incentivising vaccination itself. Misconceptions regarding what is being incentivised will need to be corrected prior to implementation to reduce concerns about coercion. Among this population, it is likely that incentives promote engagement with the target behaviour, at least in part because the value of the target behaviour is perceived to have increased.

## Supporting information

**S1 File. Questionnaire used in Study 2.**
(PDF)

## Acknowledgments

We thank Amanda Chorley and Maddie Freeman for their assistance facilitating the focus groups.

## Author Contributions

**Conceptualization:** Alice S. Forster.

**Data curation:** Lauren Rockliffe, Alice S. Forster.

**Formal analysis:** Lauren Rockliffe, Selma Stearns, Alice S. Forster.

**Funding acquisition:** Alice S. Forster.

**Investigation:** Lauren Rockliffe, Alice S. Forster.

**Methodology:** Lauren Rockliffe, Alice S. Forster.

**Project administration:** Lauren Rockliffe, Alice S. Forster.

**Supervision:** Alice S. Forster.

**Writing – original draft:** Lauren Rockliffe, Alice S. Forster.

**Writing – review & editing:** Lauren Rockliffe, Selma Stearns, Alice S. Forster.

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
