## [Decision Letter · Decision Letter 0]

7 Jul 2020

PONE-D-20-13180

A qualitative exploration of using financial incentives to improve vaccination uptake in adolescents

PLOS ONE

Dear Dr. Forster,

Thank you for submitting your manuscript to PLOS ONE. After careful consideration, we feel that it has merit but does not fully meet PLOS ONE’s publication criteria as it currently stands. Therefore, we invite you to submit a revised version of the manuscript that addresses the points raised during the review process.

While I think the exploration of using financial incentives to improve vaccination uptake in adolescents is an important topic that warrants investigation, especially in the era of anti-vaccination trends all over the world, there are several issues with both the study design, and the manuscript itself that are significant enough that they seriously undermine the contributions of the study. The manuscript has a number of weaknesses, described by both reviewers, which need to be considered.

The PLOS ONE publishes research on the basis of scientific validity and rigorous methodology. Together with the reviewers I have a number of reservations about this paper regarding both above mentioned issues. They are outlined below.

Firstly, the manuscript title does not adequately reflect the study background and should be reworded. The abstract should include some important demographic variables, i.e. setting of study, age range of the participants, and specification regarding female secondary students (instead of “adolescents”).

The Introduction section is should be shortened; some information on vaccine coverage in the country/at the study site should be provided, together with quantification of the problem caused by non-return of consent forms.

Sampling criteria should be thoroughly described, possibly with the help of graph. Table of demographics for the two studies should be included.

More detailed information of the rest of the cohort would be of value.

References are presented in a sloppy way, this should be changed following the journal criteria.

We look forward to receiving your revised manuscript.

Kind regards,

Prof. Maria Gańczak

Academic Editor

PLOS ONE

Journal Requirements:

2. Please include additional information regarding the survey or questionnaire used in the study and ensure that you have provided sufficient details that others could replicate the analyses. For instance, if you developed a questionnaire as part of this study and it is not under a copyright more restrictive than CC-BY, please include a copy, in both the original language and English, as Supporting Information

Reviewers' comments:

Reviewer's Responses to Questions

**Comments to the Author**

1. Is the manuscript technically sound, and do the data support the conclusions?

Reviewer #1: Yes

Reviewer #2: Yes

2. Has the statistical analysis been performed appropriately and rigorously? 

Reviewer #1: N/A

Reviewer #2: N/A

3. Have the authors made all data underlying the findings in their manuscript fully available?

Reviewer #1: Yes

Reviewer #2: Yes

4. Is the manuscript presented in an intelligible fashion and written in standard English?

Reviewer #1: No

Reviewer #2: Yes

5. Review Comments to the Author

Reviewer #1: Thank you very much for allowing me the opportunity to review the manuscript titled "A qualitative exploration of using financial incentives to improve vaccination uptake in adolescents" (PONE-D-20-13180). I felt confident that the authors performed careful and thorough analysis of the interesting data, however I feel like the manuscript has some errors that make it more difficult to read. I have a few comments and questions in regard to the paper; therefore I recommend that a minor revision is warranted.

Please see some specific comments below.

Title:

Authors should be more specific in title – including that the participant cohort were only female and the setting took place in London, UK. Moreover, throughout the whole manuscript, it was emphasized that incentives were used to improve consent form return rate. “A qualitative exploration of using financial incentives to improve vaccination consent form return rate in female adolescents in London, UK.”

Abstract:

Include setting of study, age range of the participants, and specify they are female secondary students instead of “adolescents” for both study 1 and 2 under methods section.

Introduction:

Line 52 – It would be informative for readers if the decline of uptake in England was quantified

Line 53 – similarly, it would also be informative for readers if the size of the 2018 measles outbreak was also quantified – how many were ill?

Line 56 – Would be clearer to read if “In the UK programmes” was clarified to “In school-based vaccination programmes in the UK...”

Line 58 - Authors said “consent forms must be returned” but in next sentence, it is said that “large proportions of consent form are not returned” which is contradicting, I would rephrasing those two sentences. It would be helpful to have confirmed that unreturned consent forms are considered consent declined.

Line 73 – extra parentheses after 8, 9 citations

Line 73 – “Most commonly they are operationalised as the offer of a reward by someone other than the target individual, although individuals may self-incentivise” It would be helpful for understanding of this sentence by providing readers of example.

Line 93 – within the cited study, was the 76% compared to 61% vaccine uptake in cohorts found as a statistically significant difference?

Line 96 – include “vaccination” between “Hepatitis B” and “consent forms”

Line 149 – change to “the objectives of the paper are” as there are two objectives

Methods and Materials

Line 159 – “In brief, participants were six secondary schools in two London boroughs” is not clear

Line 258 – “themes in Study and 2” unclear

Line 263 – Either bring up Table 1 closer to paragraph end or specify that Table 1 in Results section

Results

It would be informative to have a table of demographics for the two studies. For example, line 274 mentions 37% of whom reported to be White British ethnic background. It would be informative for readers to have more detail of the rest of the cohort, as well as the other information collected like religion, birthplace and vaccination uptake within the previous feasibility study.

Line 278 – “eight had received no doses of the vaccine” does this mean that they have declined consent to be vaccinated?

Line 324 – would not consider “commonly expressed” with 18% of responses

Table 1 – It would be useful to have a more specific table title, in case the table would stand alone. Example, to add this is from Study 2.

Table 2 – Would also be useful to specify that this is for Study 1

Line 369 – Do not understand in quotation “Get for a vaccination”

Table 3 – add in title from Study 1 quotes and Study 2 free text responses

Discussion

Line 524 – important that authors have noted that most participants in study were from families with low levels of socio-economic disadvantage – would be interesting to have study of a mixed cohort population and compare findings

References

Authors should review references list to standardize the way journal names – in some, journal names are all capitalised, some only sentence case, and some have short form of the journal

Reviewer #2: This is a well written manuscript that is easy to follow. I’m not expert in qualitative research, but the reporting is clear and framed by the COREQ recommendations.

Abstract:

Mention that the studies included 12-13 and 13-14 year old girls in London

Introduction:

Well written and informative, but appears to be too long

L 49-53 could be deleted to shorten the introduction, as the information is quite/too general

On the other hand, some information on vaccine coverage in the UK / in London / in the included neighbourhoods would be helpful, and some quantification of the problem caused by non-return of consent forms.

Methods:

Please describe how schools were selected for participation

Line 258: drop “and”

Results:

Line 284: “The most commonly reported religion was Christianity (50%) “=> rather say “Christianity was reported by half”

Was there any information about who is responsible for the form not being returned: is this really the adolescents’ choice (which the verbatims seems to suggest), or do the parents have a substantial contribution to the fact that the form is not returned (refusal to sign, loss, etc).

Discussion:

Is there evidence that the acceptability is similar among parents (incentive to adolescents)?

Which are the alternatives to incentivisation? Could the schools make the return of forms mandatory, just as other forms must be returned?

6. PLOS authors have the option to publish the peer review history of their article (what does this mean?). If published, this will include your full peer review and any attached files.

Reviewer #1: No

Reviewer #2: No

---

## [Author Response · Author response to Decision Letter 0]

16 Jul 2020

Please see attached response to reviewers comments document.

---

## [Decision Letter · Decision Letter 1]

4 Aug 2020

A qualitative exploration of using financial incentives to improve vaccination uptake via consent form return in female adolescents in London

PONE-D-20-13180R1

Dear Dr. Forster,

We’re pleased to inform you that your manuscript has been judged scientifically suitable for publication and will be formally accepted for publication once it meets all outstanding technical requirements.

Kind regards,

Prof. Maria Gańczak

Academic Editor

PLOS ONE

Additional Editor Comments (optional):

Reviewers' comments:

Reviewer's Responses to Questions

**Comments to the Author**

1. If the authors have adequately addressed your comments raised in a previous round of review and you feel that this manuscript is now acceptable for publication, you may indicate that here to bypass the “Comments to the Author” section, enter your conflict of interest statement in the “Confidential to Editor” section, and submit your "Accept" recommendation.

Reviewer #1: All comments have been addressed

2. Is the manuscript technically sound, and do the data support the conclusions?

Reviewer #1: Yes

3. Has the statistical analysis been performed appropriately and rigorously? 

Reviewer #1: Yes

4. Have the authors made all data underlying the findings in their manuscript fully available?

Reviewer #1: Yes

5. Is the manuscript presented in an intelligible fashion and written in standard English?

Reviewer #1: Yes

6. Review Comments to the Author

Reviewer #1: Thank you for responding to the reviewer comments. I believe that the authors' changes have benefited the study and the manuscript is now prepared for publication.

7. PLOS authors have the option to publish the peer review history of their article (what does this mean?). If published, this will include your full peer review and any attached files.

Reviewer #1: No

---

## [Editor Report · Acceptance letter]

10 Aug 2020

PONE-D-20-13180R1 

A qualitative exploration of using financial incentives to improve vaccination uptake via consent form return in female adolescents in London 

Dear Dr. Forster:

I’m pleased to inform you that your manuscript has been deemed suitable for publication in PLOS ONE. Congratulations! Your manuscript is now with our production department. 

If your institution or institutions have a press office, please let them know about your upcoming paper now to help maximize its impact. If they’ll be preparing press materials, please inform our press team within the next 48 hours. Your manuscript will remain under strict press embargo until 2 pm Eastern Time on the date of publication. For more information please contact onepress@plos.org.

Kind regards, 

on behalf of

Prof. Maria Gańczak 

Academic Editor

PLOS ONE